# Acontia, a Specialised Defensive Structure, Has Low Venom Complexity in *Calliactis polypus*

**DOI:** 10.3390/toxins15030218

**Published:** 2023-03-12

**Authors:** Hayden L. Smith, Peter J. Prentis, Scott E. Bryan, Raymond S. Norton, Daniel A. Broszczak

**Affiliations:** 1School of Biology and Environmental Sciences, Faculty of Science, Queensland University of Technology, Brisbane, QLD 4001, Australia; 2Centre for Agriculture and the Bioeconomy, Queensland University of Technology, Brisbane, QLD 4001, Australia; 3School of Earth and Atmospheric Sciences, Faculty of Science, Queensland University of Technology, Brisbane, QLD 4001, Australia; 4Medicinal Chemistry, Monash Institute of Pharmaceutical Sciences, Monash University, Parkville, VIC 3052, Australia; 5ARC Centre for Fragment-Based Design, Monash University, Parkville, VIC 3052, Australia; 6School of Biomedical Sciences, Faculty of Health, Queensland University of Technology, Brisbane, QLD 4001, Australia

**Keywords:** Actiniaria, acontia, phylogenetics, proteomics, toxin, venom

## Abstract

Phylum Cnidaria represents a unique group among venomous taxa, with its delivery system organised as individual organelles, known as nematocysts, heterogeneously distributed across morphological structures rather than packaged as a specialised organ. Acontia are packed with large nematocysts that are expelled from sea anemones during aggressive encounters with predatory species and are found in a limited number of species in the superfamily Metridioidea. Little is known about this specialised structure other than the commonly accepted hypothesis of its role in defence and a rudimentary understanding of its toxin content and activity. This study utilised previously published transcriptomic data and new proteomic analyses to expand this knowledge by identifying the venom profile of acontia in *Calliactis polypus*. Using mass spectrometry, we found limited toxin diversity in the proteome of acontia, with an abundance of a sodium channel toxin type I, and a novel toxin with two ShK-like domains. Additionally, genomic evidence suggests that the proposed novel toxin is ubiquitous across sea anemone lineages. Overall, the venom profile of acontia in *Calliactis polypus* and the novel toxin identified here provide the basis for future research to define the function of acontial toxins in sea anemones.

## 1. Introduction

Venoms are complex mixtures of bioactive compounds, and in phylum Cnidaria they are the primary method for assisting the capture and digestion of prey, as well as having roles in defence and intra- and inter-species competition [1,2,3,4,5]. While most venomous animals have glands and specialised delivery structures for envenomation [3], cnidarians lack a centralised venom system and, instead, have individual stinging organelles known as nematocysts throughout their body plan [4,5]. Current transcriptomic studies of toxins from sea anemones (order Actiniaria, phylum Cnidaria) have demonstrated that genes encoding their toxin arsenals are differentially expressed across tissue types and development times [1,6,7,8,9,10,11]. Subsequently, MALDI-MSI has been used to demonstrate that peptide toxins are also differentially abundant across structurally distinct tissues [10,11,12,13,14]. Previous studies have proposed that these differences in peptide toxin abundance confer different functional roles based on tissue type, for example, tentacles for prey capture, mesenterial filaments for digestion, and body column for defence [1,2].

Acontia (or acontial filaments) are a structure that is unique to specific lineages in the superfamily Metridioidea. They are formed from the mesenterial filaments [15,16] and present as long thread-like structures that are densely packed with batteries of nematocysts [16]. To date, only a few species of sea anemone are known to have acontia, with a limited number of genera reported, such as, *Actinothoe*, *Aiptasia*, *Bartholomea*, *Calliactis*, *Cereus*, *Cylista*, *Exaiptasia*, *Metridium*, and *Telmatactis* [8,15,16,17,18,19,20,21,22]. Acontia are thought to have a role in defence [15,16,19], based on studies that demonstrated reduced predation of hermit crabs when aconitate sea anemones are attached to their shell [23,24]. In addition, crude venom extracts from acontia have demonstrated cytolytic and neurotoxic activities [17,22,25], but there has been limited research into the venom composition of this structure [26]. Further hindering our understanding of the venom composition of acontia is that, in some species, it appears that acontia are transcriptionally inactive, with no detectable RNA in this structure [12]. Taken together, these data highlight the need for further investigation using a proteomic approach to expand our understanding of the protein and peptide content of acontia.

In this study, we extracted nematocysts from acontia in the sea anemone *Calliactis polypus* (Figure 1) to characterise the presence and abundance of peptide and protein toxins by using both data-dependent acquisition (DDA) and data-independent acquisition (DIA) mass spectrometry. Proteomic analysis of peptide and protein toxins in acontia from *C. polypus* revealed that the structure is dominated by a sodium channel toxin type I neurotoxin. In other species, this class of toxin is known to cause paralysis [27] or pain [28], which indicates a likely functional role in defence. Additionally, we identified a novel protein in the acontia venom that has no known functional role and a cysteine scaffold unrelated to any currently known toxins. Our results provide further insight into the functional role of acontia and identify at least one novel sequence, which further highlights the importance of using a multi-omics approach in the study of venom, particularly in highly understudied taxonomic groups.

## 2. Results

### 2.1. Acontia Sampling and Observations

Initial extraction of acontia from *C. polypus* revealed no visual presence of mucus, resulting in a clean extraction of nematocysts and low salt content.

### 2.2. Identification of Toxins in the Proteome

A total of 18 high-quality protein hits were found in venom extracted from the acontia of *C. polypus* (Table 1). Six sequences had a known toxin function in other animals, and four had significant BLAST or SMART hits to known toxin families in sea anemones. Interestingly, one sequence (c44161_g1_i1) had sequence similarity to computationally predicted proteins in sea anemones, but no known homology to or function in any other taxa. Transcript c44161_g1_i1 (hereafter referred to as ‘Unknown 12C’) contained 12 cysteine residues, suggesting a stable structure, which is integral to the function of most toxins. Furthermore, interrogation of Unknown 12C revealed a cysteine scaffold of C-X10-C-X10-C-X6-C-X3-C-X2-C-X12-C-X6-C-X11-C-X10-C-X3-C-X2-C, where Xn indicates the number of amino acids between two cysteine residues (Figure 2; Appendix A). Transcripts c50240_g1_i1 (NaTx type I), c40761_g1_i1 (Calitoxin), and c44161_g1_i1 (Unknown 12C) were the only sequences to have a high sequence coverage (>50%) of high confidence peptide matches (>0.95 confidence), with the coverage of five mature proteins having >57% (Figure 3). Additionally, NaTx type I and Calitoxin were the only candidates with very high sequence coverage of the expected mature protein. Specifically, NaTx type I had a 100% coverage and Calitoxin had a 91% coverage, assuming the conventional KR post-translational cleavage site.

Fewer putative toxins were found in the acontia of *C. polypus* than in the whole animal transcriptome. A total of 15 and 56 candidate sequences was found in the acontia proteome and whole organism transcriptome [11,29], respectively (Table 2). This represents a 27% overlap and reveals a lack of diversity and high copy number in acontia, indicating this structure may have a limited and highly specific role in envenomation.

The DIA data for the acontia of *C. polypus* showed no statistical difference between replicates. Average peak areas of high confidence peptides revealed that NaTx type I (c50240_g1_i1) was highly abundant relative to the other peptides identified. In fact, the average peak area of NaTx type I peptide matches was approximately 12–160 times higher than those of other peptides identified in the acontia of *C. polypus* (Figure 4; Appendix A). The average peak area of the second and third most abundant matches were for Unknown 12C (c44161_g1_i1) and PLA2 (c56806_g1_i1), which were 12 and 14 times less abundant than NaTx type I, respectively. This further indicates that the acontia has limited toxin diversity and, thus, probably a highly specific role in envenomation.

### 2.3. Comparison of Five Toxins of Interest in Sea Anemone Species

Overall, there was a greater variation in transcript copy number for the NaTx type I and sea anemone 8 toxin groups, despite the identification of contamination in four of the 14 species investigated (Table 3; Appendix A). NaTx type I had a copy number range between zero and six and was only found in five species, while sea anemone 8 had a range between two and nine and was found in all species except *A. plumosum*. Two of the other three toxin families (PLA2 and the Unknown 12C) were found in the transcriptome assemblies of all sea anemone species and had copy number ranges of 0–3, 1–3, and 1–2 for KTx type III, PLA2, and the Unknown 12C, respectively.

### 2.4. Phylogenetic Tree Analysis

Phylogenetic analysis of the Unknown 12C sequences was resolved in a tree with four well-supported clades (Figure 5). Clade one contained only two sequences from species of the superfamily Edwardsioidea and was sister to the other three clades. Clades two and four were both Metridioidea-specific clades, with both clades containing one gene sequence from each Metridioidea species. Clade three was an Actinioidea-specific clade containing one or two sequences from each species. The Unknown 12C toxin candidate was found in clade four and arose through a Metridioidea-specific gene duplication event. Interestingly, an Unknown 12C sequence identified in the milked venom proteome of *T. stephensoni* (TR3057|c0_g1_i1) [12] was found in Metridioidea-specific clade two, which does not contain the *C. polypus* acontia sequence, supporting the possibility that both clades contain peptides with toxin function.

### 2.5. Selection Analysis for Unknown 12C

Selection analysis of the Unknown 12C gene showed evidence of purifying selection. A total of 52 of 175 sites had a dN/dS ratio of less than 0.2 with a *p*-value ≤ 0.05, and all codons that encode cysteine residues had a value of zero (Figure 6; Appendix A). Additionally, there was no evidence to support diversifying selection at either the branch or site levels.

## 3. Discussion

The study of sea anemone venom has increased in recent years [4,5,9,30,31,32,33,34,35,36], but of note is the lack of proteomic research to understand venom composition for most of the specialised envenomation structures in sea anemones. To gain a better understanding of venom composition in a specialised envenomation structure, we undertook the first proteomic study to identify the protein and peptide profile of toxins present in the venom from the acontia of *C. polypus*.

### 3.1. Limited Toxin Diversity in Acontia Venom Supports the Dominant Toxin Hypothesis

Acontia are a highly specialised envenomation structure laden with batteries of venom-containing nematocysts that are found only in specific lineages of the superfamily Metridioidea [16,37]. The venom extracted from *C. polypus* acontia analysed in this study was found to have low complexity, containing a limited number of protein and peptide toxin families. Furthermore, acontia venom was characterised by low copy numbers in toxin families, a trait that is seen in the milked venom of some other sea anemones [12,31], but the *C. polypus* acontia venom had markedly lower venom complexity. In fact, a single dominant toxin, NaTx type I, made up most of the venom. This indicates that the acontia of *C. polypus* delivers a high dose of a single toxin rather than a concoction of toxins. The over-representation of a single NaTx is in strong agreement with an ancestral state reconstruction analysis of toxin gene expression that found NaTx to be the dominant toxin expressed in the Metridioidea species analysed, which included *C. polypus* [38]. The low complexity of the *C. polypus* acontia venom confirms the validity of the ‘dominant toxin hypothesis’, where a single toxin or toxin family dominates the venom phenotype in sea anemones [38].

### 3.2. Venom in the Acontia Supports a Defensive Role

Acontia is expelled as a defensive response when anemones are contacted by invertebrate predators. The venom present in the genus *Calliactis* has been shown to strongly deter invertebrate predators, including crustaceans and octopuses [23,24], with the release of acontia vastly reducing predation from crustaceans [24]. The identification of known neurotoxins (NaTx and KTx) and enzymes (proteases and PLA2) in the acontia venom is consistent with a role in defence by deterring predators through the induction of lesions, pain, or paralysis. In fact, the presence of both neurotoxins and enzymes validates previous research that crude venom derived from acontia has both cytolytic and neurotoxic activity [17,22,25] and that specific fractions of this venom cause paralysis in crustaceans [39]. However, further research is required to better understand if the neurotoxins identified in the acontia of *C. polypus* could be responsible for the deterrence of crustaceans during predatory attacks of sea anemones [24].

The functionality of toxins found in acontia venom also supports its role as a defensive structure. The dominant toxin found in acontia, NaTx type I, is a highly studied neurotoxin in sea anemone species and has been shown to have a broad spectrum of sodium channel activation across the Nav1.1-1.6 channels, inducing both paralysis and pain in different models [28,33,40]. Similarly, ShK toxins are well-studied neurotoxins that predominantly inhibit voltage-gated potassium channels, with some ShK toxins having a broad range of activity across the Kv1.1-1.6 channels [41,42,43]. Furthermore, Calitoxin is a potent neurotoxin, which causes massive neurotransmitter release and repetitive firing of the axons in crustaceans [39]. Phospholipase A2, the cytolytic toxin found in highest abundance, has been widely studied across multiple animal groups and has an important role in membrane attachment and disruption in sea anemone venom [25,44]. The presence of these well-studied toxins in acontia venom further reinforces that acontia venom can cause the induction of lesions, pain, or paralysis, and further highlights the need for proteomic and functional research to elucidate the role of the toxins identified in sea anemones.

### 3.3. Discovery of a Novel Peptide in Acontia

Novel genes and peptides represent an opportunity to gain greater insight into the function of a structure in an organism; in this respect, Unknown 12C is an ideal candidate for future study. This Unknown 12C sequence was found in the acontia proteome of *C. polypus*, and a paralogous 12C sequence was identified in the milked venom proteome of *T. stephensoni* [12]. The presence of this cysteine-rich sequence in the venom of two separate species strongly suggests that it is a functional protein, most likely with toxin activity. Similar to many sea anemone peptide toxins [11,38], multiple codons from the Unknown 12C protein were found to be evolving in a manner consistent with the action of negative selection, which was most prominent for the cysteine encoding codons. The cysteine framework has no similarity to any known toxin in sea anemones [45,46], although it has an identical motif at two locations in the scaffold that match the last three C-X frames of ShK toxins (C-X3-C-X2-C) [47] and sea anemone 8 toxins [48]. This scaffold, as well the presence of the Lys-Arg (KR) motif at location 71–72 of the protein sequence (Figure 2), may indicate that Unknown 12C has two tandem domains with a scaffold of six cysteines each, and that it is similar to but divergent from ShK toxins. The KR motif is only present in one other sea anemone analysed, *N. annamensis* (transcript TR35413|c1_g1_i1; Appendix A), and all other sequences lack the KR motif for the mature peptide cleavage site that is typical of neurotoxins [49,50]. This suggests that there is a greater likelihood that it has a 12-cysteine scaffold instead of two 6-cysteine domains. The structure and function of the Unknown 12C protein in sea anemones needs to be elucidated, but its unusual sequence structure and presence in all sea anemones analysed in this study indicate that it may be a novel toxin found in a wide range of sea anemone species.

## 4. Materials and Methods

### 4.1. Sea Anemone Collection and Acontia Sampling

Five specimens of *C. polypus* (Figure 1) were obtained from sea-rafted pumice at Frenchman’s Beach, North Stradbroke Island (Queensland, Australia; 27°25’44.21” S 153°32’36.30” E). Following collection, the anemones were acclimated in an artificial sea water (ASW) system at Queensland University of Technology, Brisbane. Anemones were fed artemia and then deprived of food for three days before extracting acontia. Sea anemones were rinsed with fresh ASW before being mechanically agitated to release acontia. Acontia was washed with ASW, incubated in 1 M sodium citrate for 10 min, and then briefly pipetted to release nematocysts in solution, before centrifuging at 5000× *g* for 1 min. Supernatant was discarded and nematocysts stored at −20 °C. Four biological replicates, including three technical replicates of each, were acquired.

### 4.2. Proteome Extraction and Generation

Nematocysts were resuspended in Milli-Q water to allow osmotic pressure to release peptide and protein toxins from organelles, followed by sonication in a water bath (LGO model 60W-18CH) for 10 min at 40 KHz to disrupt any nucleic acids and release any unstimulated nematocysts. Protein concentration was estimated by absorbance at 280 nm using a Nanodrop, and approximately 20 µg of protein was subsequently reduced and alkylated using a previously reported method [51] with the following changes: diluted with 50 mM ammonium bicarbonate to a final volume of 49.5 µL, reduced by the addition of 0.5 µL of 1 M dithiothreitol, and alkylated with 2 µL of 1 M iodoacetamide. Digested samples were desalted using an SCX resin Stage-Tip approach [52] before being resuspended in 1% formic acid containing indexed retention time (iRT) peptides. An aliquot from all peptide samples was pooled and this sample was analysed by DDA to determine the 100 optimal SWATH-MS variable windows (minimum width of 3 Da and overlap of 1 Da) for DIA analysis for each individual sample. Data were acquired on a TripleTOF 5600+ mass spectrometer (Sciex) coupled to an Eksigent microflow liquid chromatography system using an increasing gradient of 3–80% of Solution B (99.9% acetonitrile in 0.1% formic acid) in 0.1% formic acid over 80 min.

### 4.3. Proteome Annotation

A previously assembled transcriptome for *C. polypus* [11] was used as a search database (fasta formatted) for the mass spectra. The transcriptome was filtered for toxin and toxin-like candidates using Blast+ (v2.9.0) against a database derived from UniProt using the following query syntax: toxin OR annotation: (type: “tissue specificity” venom) AND reviewed: yes (accessed on 10 November 2021). Mass spectra searches were completed using ProteinPilot (v5.0) following the standard search parameters from previous publications for sea anemone venoms: trypsin digestion; iodoacetamide cysteine modification; biological modifications and amino acid substitutions; confidence threshold of 0.05 with false discovery rate analysis [11,45]. SWATH mass spectra were processed using Skyline (v21.2) following the standard analysis parameters, including: trypsin digestion; cysteine carbamidomethyl modification; precursor charges two–four; ion types y, b; MS/MS filtering using DIA acquisition method, DDA custom window isolation scheme, and centroided mass analyser at 10 ppm. Relative abundances were determined after normalisation of data using iRT retention and equalised medians. MSstats was used to compare biological replicates for acontia samples using Skyline’s Group Comparison parametric test for statistical significance (q-value < 0.05). Raw data have been deposited to the ProteomeXchange Consortium via the PRIDE partner repository (https://www.ebi.ac.uk/pride/) with the dataset identifier PXD037063.

### 4.4. Toxin and Toxin-like Gene Identification

Previously assembled transcriptomes (*Actinia tenebrosa*, *Actinodendron plumosum*, *Anemonia sulcata*, *Anthopleura buddemeieri*, *Aulactinia veratra*, *C. polypus*, *Edwardsiella carnea*, *Exaiptasia diaphana*, *Megalactis griffithsi*, *Nemanthus annamensis*, *Nematostella vectensis*, *Stichodactyla mertensii*, *Telmatactis stephensoni*, and *Triactis producta*) [11,29] were used to identify homologous candidates for the genes encoding five toxin peptides and proteins found in the acontia proteome of *C. polypus*: sea anemone potassium channel inhibitor (KTx) type III, sea anemone sodium channel inhibitor (NaTx) type I, phospholipase A2 (PLA2), sea anemone 8, and an unknown cysteine-rich protein. Candidates were manually filtered to ensure that sequences were full-length ORFs, had a signal peptide, and did not have identical sequences due to contamination of the sequencing platform. Additionally, protein domains were verified using SMART (http://smart.embl-heidelberg.de/) [53] for phospholipase A2 homologous sequences, and conserved motifs/residues were verified for all homologous sequences to ensure integrity of the data obtained. Additionally, *C. polypus* sequences were used as blastx queries against the UniProt database (https://www.uniprot.org/) to obtain homologous sequences from the Swissprot/Reviewed database. Multiple identical transcripts from the NaTx type I and sea anemone 8 toxin gene families were identified in the transcriptomes of *A. plumosum*, *S. mertensii*, *T. stephensoni*, and *T. producta* (NCBI accessions: SRX7189368, SRX6886104, SRX1643233, SRX5112246, respectively). These sequences are likely the result of index hopping and were excluded from all downstream analyses using the parameters previously set out in Surm et al. [11].

### 4.5. Phylogenetic Analyses

A phylogenetic tree for the Unknown 12C toxin protein was constructed from the amino acid sequences for all verified sequences. All sequences were trimmed to remove signal peptides. Sequences were aligned using MUSCLE [54] and trees were constructed using IQ-TREE (http://iqtree.cibiv.univie.ac.at/) [55]. The best-fit model was determined automatically by IQ-TREE and maximum-likelihood trees were generated using 10,000 ultrafast bootstrap alignments and Bayesian-like approximate likelihood ratio testing [56].

### 4.6. Selection Analyses

Selection analysis was performed on the Unknown 12C gene of interest to determine if selection was acting on this potential peptide toxin. All nucleotide sequences identified as Unknown 12C were extracted from the transcriptomes and trimmed to remove signal peptides. Sequences were aligned using MUSCLE [55], and the alignment analysed using Mixed Effects Model of Evolution [57] (*p*-value ≤ 0.05) and Fixed Effects Likelihood [58] (100 bootstrap resampling *p*-value ≤ 0.05) via the Datamonkey server (http://www.datamonkey.org/) to test if individual sites have been subjected to selection pressure.

## Figures and Tables

**Figure 1 toxins-15-00218-f001:**
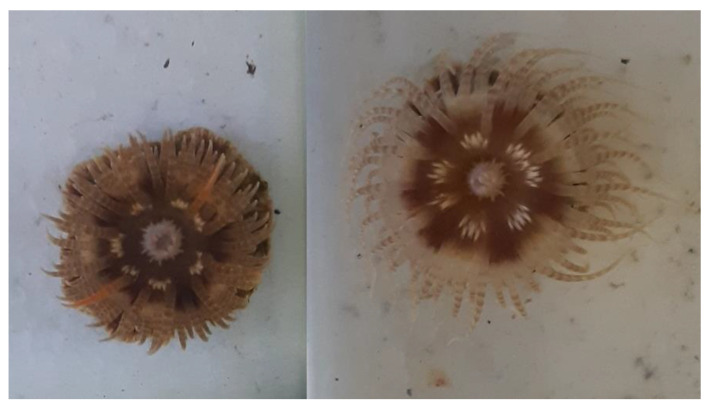
Cropped images of *Calliactis polypus* in artificial sea water tanks.

**Figure 2 toxins-15-00218-f002:**
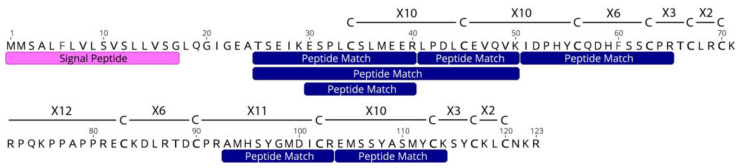
Amino acid sequence of Unknown 12C (transcript c44161_g1_i1) depicting the cysteine framework, signal peptide, and peptide matches from the mass spectra of the SWATH-MS and DDA pooled acontia sample. Xn linkages represent the number of amino acid residues connecting cysteine residues. Proposed pre-peptide cleavage fragment (LQGIGEA) is located between the signal peptide and the first peptide match.

**Figure 3 toxins-15-00218-f003:**
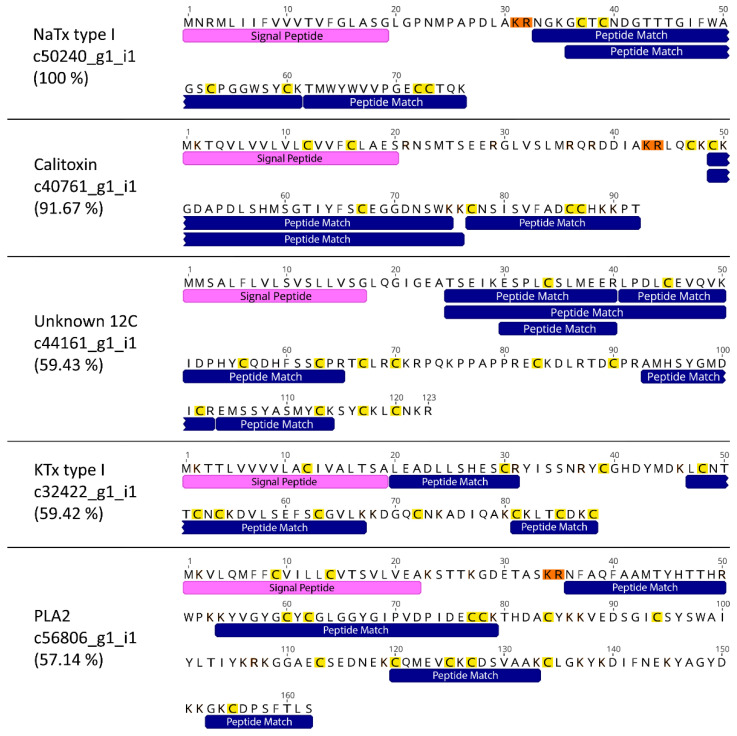
Amino acid sequences of the five toxins with the highest percent coverage for their mature proteins. Sequences depicted with signal peptides and peptide matches from the mass spectra of the SWATH-MS and DDA pooled acontia sample. Post-translational cleavage sites (KR) are highlighted in orange and cysteine residues are highlighted in yellow.

**Figure 4 toxins-15-00218-f004:**
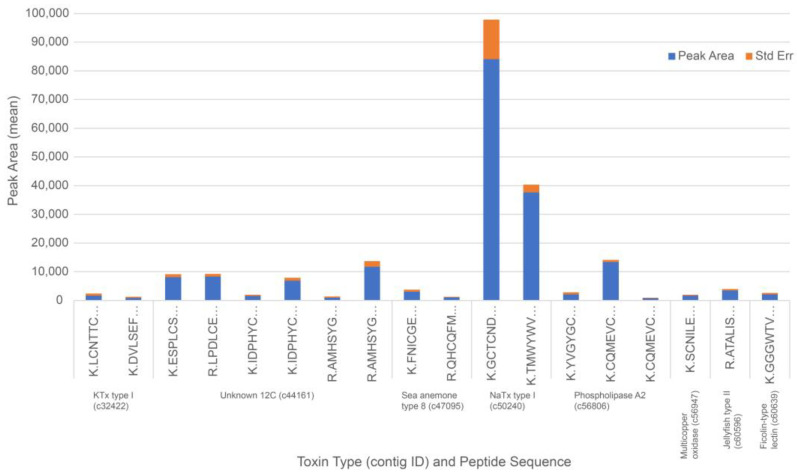
Relative abundance of peptide matches for putative toxin candidates found in the mass spectra of acontia. Full notation of toxin sequences and peptide matches can be found in Appendix A.

**Figure 5 toxins-15-00218-f005:**
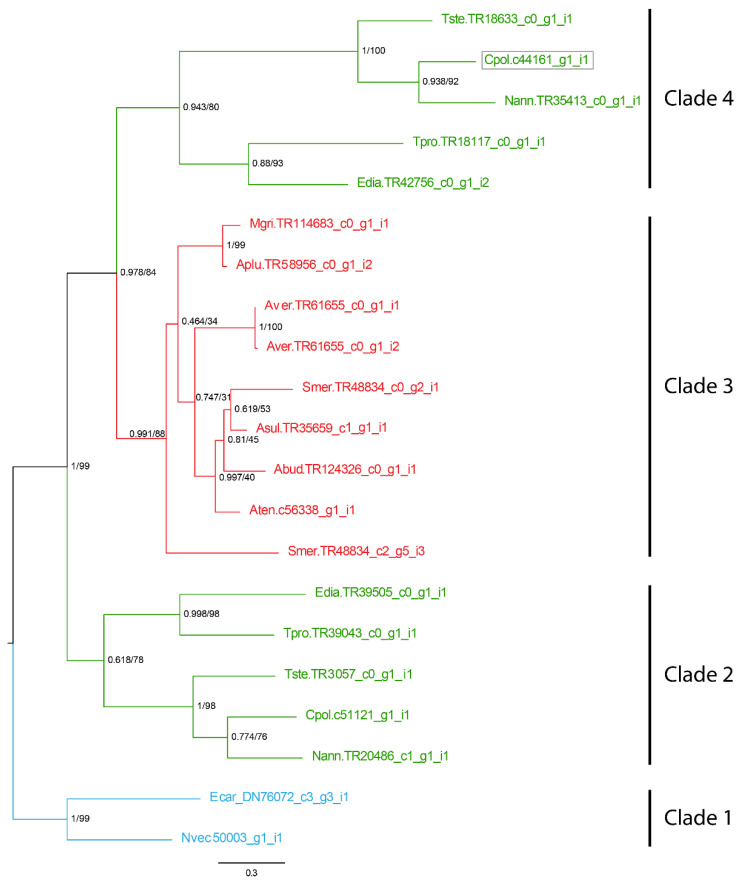
Maximum-likelihood phylogenetic tree of Unknown 12C, in the transcriptomes of actiniarian species, with maximum-likelihood bootstrap support (ML) value and Bayesian-like approximate likelihood ratio testing (aLRT) value. Sequence found in the proteome of acontia highlighted in grey box. Actinioidea branches and sequences highlighted in red, Metridioidea branches and sequences highlighted in green, and Edwardsioidea branches and sequences highlighted in blue. Support values shown as aLRT (0–1)/ML bootstrap (0–100). Gene identities were given a nomenclature as follows: Cpol_c44161_g1_i1—C: first letter of genus; pol: first three letters of species followed by an underscore; c44161_g1_i1: transcript ID as per the transcriptome.

**Figure 6 toxins-15-00218-f006:**
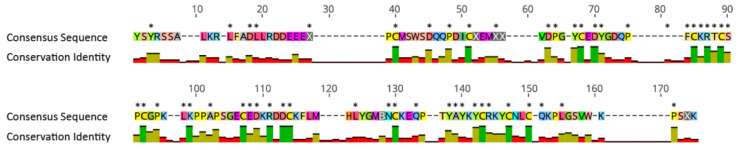
Consensus amino acid sequence for the alignment of the Unknown 12C gene. Conservation identity denotes percentage of sequences that match the consensus sequence (green bar for 100%; yellow bar for ≥50% and <100%; red bar for <50%). Asterisk denotes sites under purifying selection (dN/dS ratio < 0.2).

**Table 1 toxins-15-00218-t001:** Putative toxin candidates with significant peptide matches from mass spectra of the SWATH-MS and DDA pooled acontia sample. Toxin families with a known function are highlighted in bold and sea anemone toxins are highlighted in italics. Sequence coverage refers to whole protein sequence and mature protein coverage refers to sequence without predicted signal peptide and cleavage sites.

Toxin Family	Contig	Peptide Hits (Conf ≥ 95)	Sequence Coverage (%)	Mature Protein Coverage (%)
Disintegrin and Metalloproteinase	c59360_g1_i1	15	14.69	14.96
Disintegrin and Metalloproteinase	c62072_g1_i1	10	13.07	13.40
Ficolin-type lectin	c60639_g1_i1	3	10.42	11.11
**Jellyfish type II**	c60596_g1_i1	4	7.60	7.83
*KTx type I (ShK)*	c32422_g1_i1	4	46.59	59.42
*KTx type I (ShK)*	c50551_g1_i1	3	42.16	52.44
**KTx type III**	c8939_g1_i1	2	26.83	46.81
**Multicopper oxidase**	c56947_g1_i1	5	21.80	23.39
*NaTx (Calitoxin)*	c40761_g1_i1	4	52.17	91.67
**NaTx type I**	c50240_g1_i1	5	57.89	100.00
Peptidase M12A	c26461_g1_i1	7	11.97	12.40
Peptidase M12A	c49341_g1_i3	4	2.47	2.55
**Peptidase M13**	c63393_g1_i1	13	5.83	5.99
**Phospholipase A2**	c56806_g1_i1	7	49.38	57.14
*Sea anemone type 8*	c47095_g1_i1	2	24.39	31.75
Unknown	c44161_g1_i1	7	51.22	59.43
Unknown	c56815_g1_i2	6	37.65	42.66
Unknown	c58770_g1_i1	17	47.61	50.56

**Table 2 toxins-15-00218-t002:** Putative toxin candidates found in the transcriptome and venom of *C. polypus*, according to toxin function and family groups. Transcript copy number refers to number of contig IDs that have significant matches to the NCBI, PFAM, and/or SMART databases for known toxin functions. Transcripts observed at peptide level refer to the number of translated transcripts that have high confidence peptide matches to the acquired MS spectra.

Functional Category	Toxin Family	Toxin Subtype	Transcript Copy Number	Transcripts Observed at Peptide Level
Enzyme	Lectin	C-Type	4	0
Enzyme	Lectin	Ficolin	2	1
Enzyme	Lipase	AB hydrolase	2	0
Enzyme	Lipase	Phospholipase A2	4	1
Enzyme	Lipase	Type B carboxylesterase	1	0
Enzyme	Metalloprotease	Disintegrin and metalloprotease	2	2
Enzyme	Metalloprotease	Peptidase M12A	8	2
Enzyme	Protease	Multicopper oxidase	4	1
Neurotoxin	Potassium channel toxin	Kazal	2	0
Neurotoxin	Potassium channel toxin	Type I (ShK)	4	2
Neurotoxin	Potassium channel toxin	Type II (venom kunitz)	5	0
Neurotoxin	Potassium channel toxin	Type III	1	1
Neurotoxin	Sodium channel toxin	Calitoxin	1	1
Neurotoxin	Sodium channel toxin	Sea anemone sodium channel toxin	2	1
Neurotoxin	Sodium channel toxin	Type I	1	1
Unknown	Structural class peptide	Sea anemone type 8	6	1
Unknown	Structural class peptide	Sea anemone type 9	1	0
Unknown	Unknown	Unknown (12C)	1	1
Unknown	Unknown	Cephalotoxin	1	0
Unknown	Unknown	VP302	4	0
		**Total**	**56**	**15**

**Table 3 toxins-15-00218-t003:** Transcript copy numbers for five toxin genes of interest in three superfamilies of sea anemones. Transcripts obtained from published transcriptomes by Surm et al. [11] and van der Burg et al. [29].

Superfamily	Species	Transcript Copy Number				
		KTx Type III	NaTx type I	PLA2 (12C)	Sea Anemone 8	Unknown 12C
Actinioidea	*Actinia tenebrosa*	1	1	1	6	1
Actinioidea	*Actinodendron plumosum*	0	0	1	0	1
Actinioidea	*Anemonia sulcata*	3	1	1	6	1
Actinioidea	*Anthopleura buddemeieri*	2	1	1	5	1
Actinioidea	*Aulactinia veratra*	1	0	1	6	2
Actinioidea	*Megalactis griffithsi*	0	0	1	2	1
Actinioidea	*Stichodactyla mertensii*	1	0	3	2	2
Edwardsioidea	*Edwardsiella carnea*	0	0	1	4	1
Edwardsioidea	*Nematostella vectensis*	0	0	1	3	1
Metridioidea	*Calliactis polypus*	1	3	4	9	2
Metridioidea	*Exaiptasia diaphana*	1	0	2	4	2
Metridioidea	*Nemanthus annamensis*	1	6	2	5	2
Metridioidea	*Telmatactis stephensoni*	2	0	3	7	2
Metridioidea	*Triactis producta*	2	0	1	3	2
	**Total**	**15**	**12**	**23**	**62**	**21**

## Data Availability

Raw data have been deposited to the ProteomeXchange Consortium via the PRIDE partner repository (https://www.ebi.ac.uk/pride/) with the dataset identifier PXD037063.

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
