# Peer review of "Acontia, a Specialised Defensive Structure, Has Low Venom Complexity in Calliactis polypus"

_toxins, 2023, doi:10.3390/toxins15030218_

Round 1
Reviewer 1 Report
The authors have written an incorrect abstract and mislead readers about article content. The work is devoted to proteomics research of the venom sample based/confirmed the transcriptomics data published previously. It is interesting to know how many previously predicted toxins were found, the predominance of one component over others toxins, targets specificity of compounds confirmed by MS. Is a quantity proportion between venom compounds by transcriptome same to proteome? These points necessary to reflect in the abstract. Additionally, authors can to discuss results of evolutionary comparisons, but the morphological features and description of the species should be shortened, since this article pay low attention to them.
The article corresponds to the journal scope, but have interest for a narrow specialists, for whom it is necessary to write an informative abstract.
Based on the results presented, I am interested in the opinion of the authors about Unknown 12C (Figure 2 and Supplementary Figure 1). It seems to me that the gene codes sequences of two toxins, but the site of their processing by maturation is different from the classic proprotein convertase. I would like to get the opinion of the authors about this.
Reviewer 2 Report
The reference paper is a good contribution to the knowledge of sea anemone toxins, particularly those presents in the acontia. Is well written and the findings are supported by the obtained results. The main findings are the identification of a novel protein in the acontia venom that has no known functional role and a cysteine scaffold unrelated to any currently known toxins; and the confirmation of the validity of the ‘Dominant toxin hypothesis’, where a single toxin or toxin family dominates the venom phenotype in sea anemones due to the low complexity of the C. polypus acontia venom.
I have minor concerns detailed here:
Lines 34, 282. Acontia are not a tissue. The tissues present in sea anemones are the ectoderm and endoderm. The acontia, rather than a tissue, are an structure.
Line 37. I understand that the authors did not intend to add an exhaustive list of genus here. There are other genera with acontia not mentioned here (eg. Anthothoe, Tricnidactis, Diadumene, etc.), to mention a few more.
Line 157. It is not clear if the size range provided (12-25 µm) involves the two types, please clarify and add a range for each type. Also could be nice if the mastigophores are more specifically identified.
